# Biochemical and Behavioural Alterations Induced by Arsenic and Temperature in *Hediste diversicolor* of Different Growth Stages

**DOI:** 10.3390/ijerph192315426

**Published:** 2022-11-22

**Authors:** Pedro Valente, Paulo Cardoso, Valéria Giménez, Marta Sofia Salvador Silva, Carina Sá, Etelvina Figueira, Adília Pires

**Affiliations:** 1Department of Biology, University of Aveiro, 3810-193 Aveiro, Portugal; 2Department of Biology & CESAM—Centre for Environmental and Marine Studies, University of Aveiro, 3810-193 Aveiro, Portugal

**Keywords:** metalloids, global warming, invertebrates, behaviour, oxidative stress

## Abstract

Contamination with Arsenic, a toxic metalloid, is increasing in the marine environment. Additionally, global warming can alter metalloids toxicity. Polychaetes are key species in marine environments. By mobilizing sediments, they play vital roles in nutrient and element (including contaminants) cycles. Most studies with marine invertebrates focus on the effects of metalloids on either adults or larvae. Here, we bring information on the effects of temperature increase and arsenic contamination on the polychaete *Hediste diversicolor* in different growth stages and water temperatures. Feeding activity and biochemical responses—cholinesterase activity, indicators of cell damage, antioxidant and biotransformation enzymes and metabolic capacity—were evaluated. Temperature rise combined with As imposed alterations on feeding activity and biochemical endpoints at different growth stages. Small organisms have their antioxidant enzymes increased, avoiding lipid damage. However, larger organisms are the most affected class due to the inhibition of superoxide dismutase, which results in protein damage. Oxidative damage was observed on smaller and larger organisms exposed to As and temperature of 21 °C, demonstrating higher sensibility to the combination of temperature rise and As. The observed alterations may have ecological consequences, affecting the cycle of nutrients, sediment oxygenation and the food chain that depends on the bioturbation of this polychaete.

## 1. Introduction

Coastal systems often serve as sinks for pollutants arising from aquaculture, shipping, agriculture, industry, mining and sewage treatment plants [1], causing environmental decay of natural conditions [2,3]. Metals and metalloids toxicity varies according to the element but are usually toxic at sufficiently high levels. The difference between optimal and toxic levels depends on the physiological needs of organisms [4]. Some metals are naturally occurring and are biologically essential (e.g., copper, chromium, manganese and zinc, which are essential for organisms’ growth and life cycles). However, high concentrations of these elements can cause toxicity [4,5]. Others are non-essential metals (e.g., cadmium, lead and mercury), since they do not have any function for organisms whose accumulation is dangerous for their living being [6,7,8]. Metalloids are semimetals that share a physical appearance with metals but act chemically like nonmetals [7] and are dangerous even in lower concentrations [7,9]. Natural sources of metals and metalloids include minerals by disintegration and alteration of minerals and rocks by physical-biogeochemical processes, volcanic eruptions, forest fires and biogenic sources [5,10,11]. However, they also have anthropogenic sources like coal and oil combustion, metal mining, smelting and refining, fertilizers, pesticides and waste incineration [11]. Urbanization of marine coastal areas is also contributing to the increase in these contaminants in estuarine coastal systems [12].

Arsenic (As) is a toxic metalloid that is among the most found inorganic contaminant and often occurs from natural sources (e.g., earth crust) as arsenates, sulfides, sulfosalts, arsenides, arsenites, oxides, silicates and elemental As [13]. Regarding this, all organisms are constantly in contact with this element by air, water consumption and food ingestion, which, when natural occurring, humans do not accumulate more than 4 mg of As and marine animals 0.3 mg [13,14]. However, as a result of human activities like the fertilizer industry and mining activity, As concentration on aquatic ecosystems has increased [15,16]. Previous studies demonstrated that this contaminant tends to bioaccumulate more on producers (e.g., algae) and then to biomagnify to first consumers (e.g., crustaceans, bivalves, annelids) [16]. However, due to the detoxification mechanisms of some organisms (e.g., phytoplankton, bacteria, etc.) that transforms arsenic inorganic forms to methylated and organic forms, decreasing this biomagnification with the increase in trophic levels [16]. Exposure to As can induce blood pressure alteration in pregnant women [17], cause cancer in the urinary bladder, lungs and skin in humans [18] and can induce infertility by reducing sperm production, number and quality in rats [19]. Previous studies observed that As can accumulate in fish tissues that were fed a diet contaminated with metals and As [20,21], affecting their growth [21] and reproduction [20]. Other studies have reported the effects of As on important endpoints in polychaetes, such as regenerative capacity, behaviour, antioxidant defences, and oxidative damage [22,23,24]. These studies demonstrated that polychaetes exposed to As needed more time to fully regenerate and regenerated fewer segments [22,24] and had higher levels of oxidative stress [22,23,24].

Alongside pollution, global warming is also a worldwide threat, contributing to water acidification, and temperature and seawater salinity increases [25]. The rising temperature can be experienced throughout almost the entire globe. In the decade of 2006–2015 was observed the increase in global mean surface temperature by 0.87 °C, compared to the last century [25]. By the end of the century, it is estimated that global mean surface temperature will increase between 0.95 and 2.0 °C depending on how successful is the reduction of CO_2_ emission [25]. Previous studies demonstrated that temperature rise impacts marine organisms, by increasing diseases and mortality in shrimps [26], redistribution of species, specially fish, to higher latitudes, contributing to homogenization of marine fauna in lower latitudes [27,28,29], decreasing burrowing activity and delay on the regenerative capacity of polychaetes [30,31], larval increase, otolith and somatic growth on northern pike [32], increasing of cannibalism behaviour at early age due to length difference between on *Esox lucius* L. [33]. In addition, it is also expected that the temperature increase modulates the susceptibility of organisms to pollutants through alterations in the rate of biochemical and physiological processes, but may also change pollutants bioavailability and toxicity [34,35].

Polychaetes are usually the most abundant taxonomic group in estuarine environments [36,37] and play an important role in food chains being a valuable food source to crustaceans, birds and fish populations [38,39,40]. Additionally, these organisms can accumulate large amounts of contaminants, as they live in close contact with the sediment and pore water [41], and by their various feeding strategies like suspension-feeding, deposit-feeding and omnivorism [42]. Several studies demonstrated that these marine species are good bioindicators of contaminants, such as metals and metalloids [41], nanoplastics [43] and polycyclic aromatic hydrocarbons [38], among others. Moreover, the activities of macrobenthic polychaetes have significant impacts on sediments mobilization and organisms, like the polychaeta *Hediste diversicolor* (O.F. Müller, 1776). In fact, the deposition of these contaminants in sediments is a natural remediation that removes them from circulating from the environment and accumulating in the sediments [44]. Environmental studies conducted in the Ria de Aveiro Lagoon (Portugal), demonstrated the capacity of the polychaete *Diopatra neapolitana* to accumulate various metals (Cr, Ni, Cu, Pb, Cd, Hg) and As, leading to cellular damage and increased antioxidant and biotransformation enzymes activity [24]. Moreover, the same study demonstrated that the accumulation of the studied metals and As affected the regenerative capacity of *D*. *neapolitana*, in which organisms regenerated fewer segments and took longer to regenerate. Under laboratory conditions exposure of *Perinereis aibuhitensis* to Cd to for 8 days suggested that Cd interfered with the antioxidant defense system of this polychaete species [45]. Bouraoui et al. [46] also observed the induction of oxidative stress biomarkers in different body regions of the polychaete *H. diversicolor* exposed to Cu.

*H. diversicolor*, known as common ragworm, is a polychaete species belonging to the phylum Annelida, family Nereididae. Polychaetes nereidids are characterized by the presence of paragnaths, known as chitinous denticles, distributed in groups on maxillary belts and oral belts of the pharynx [47]. The species *H*. *diversicolor* is characterized by an eversible proboscis with paragnaths on oral and maxillary belts, a subtriangular prostomium with four small eyes, two large palpi and two short frontal antennae, a peristomium with four pairs of tentacular cirri and a length of 45–92 segments [48,49]. The paragnaths located on maxillary belts have the function to grab the food and transfer it towards the gut by pharynx retraction [50]. The paragnaths located on oral belts have the function to burrow and browse on the sediment [51]. This species inhabits the shallow marine and brackish waters, in mud sand but also gravels, clays and turf, distributed in the North temperature zone from both the European and the north American coast of the Atlantic [49]. *H*. *diversicolor* has great ecological tolerance, being found in European estuaries, where it plays an important role in these ecosystems [49]. It creates borrows causing bioturbation, which potentializes sediment oxygenation, affecting the cycle of nutrients and the availability of contaminants [49,52]. The common ragworm has a commercial interest because it is used for recreational fishing and food in aquaculture [49].

Although several studies on the toxicological effect of As and temperature increase on marine invertebrates have been reported in the literature, there is no data in the literature that allows us to conclude that they may have an effect according to organism’s size, particularly on polychaetes. Therefore, the aim of this study is to evaluate the chronic effects of As on *H*. *diversicolor* with different sizes and at different temperatures (16 °C and 21 °C). The selected parameters used to evaluate these effects were the activities of the enzyme’s cholinesterase, glutathione S-transferases, catalase and superoxide dismutase, lipid peroxidation and protein carbonylation to quantify the extent of oxidative damage, and energy related parameters and feeding activity.

## 2. Materials and Methods

### 2.1. Test Organisms

Specimens of *H*. *diversicolor* were born in the laboratory [23] and maintained in glass aquaria with artificial seawater (salinity 28 and pH 7.8) and sediment (at a ratio 3:1) in a temperature-controlled room (16 ± 1 °C), under continuous aeration. During this nursery time, organisms were fed *ad libitum* with commercial fish food every 3/4 day (Protein 46.2%, Fat 8.9%). Water was renewed twice a month.

### 2.2. Experimental Design

Organisms were separated by size (1–2.5 cm (small, with an age of 3 months), 3–5 cm (medium, corresponding to an age of 5 months), 6–9 cm (large, organisms with 7 months)) and then, organisms of each size (20 small, 10 medium and 5 large × 3 replicates), were exposed for 28 days to As (0, 0.05, 0.25 mg/L) at two temperatures: 16 °C and 21 °C, under continuous aeration. For each condition, aquariums were filled with sand and artificial sea water (1:2). To avoid a temperature shock that would happen if the worms were distributed in the first day in each aquarium with the desired temperature, every aquarium started with the same temperature, at the control 16 °C. Every day, the temperature was increased 1 °C. After reached the desired temperature, polychaetes were exposed to the selected conditions for 28 days. A stock solution of sodium arsenate (Na3AsO4) (CAS no. 10048-95-0, Sigma-Aldrich, St. Louis, MO, USA) was prepared in ultrapure water and spiked in aquaria to reach nominal Arsenic concentrations of 50 and 250 μg/L. Arsenic concentrations were chosen according to the literature available for Ria de Aveiro, Portugal, and for other contaminated areas [24,53,54]. The range of As concentrations found in natural waters varies from 0.5 μg/L to more than 5 mg/L [54]. At Ria de Aveiro, the range of As concentrations found in sediments vary from 0.82 mg/kg to 94 mg/kg [24,53].

Temperature conditions were tested based on values recorded at Ria de Aveiro and possible climate change scenarios predicted to occur in the near future [25,31,55]. Temperature conditions tested were selected as representative of mean values measured at the sampling site (16 °C), and taking in account the future projections (21 °C) estimated an average surface temperature increase between 2.6 °C and 4.8 °C by the end of this century [25] and Viceto et al. [56] indicated that the projected temperatures for Iberian Peninsula show an increase of over 6 °C by 2081–2100.

Water was renewed every week, at the appropriate temperature, to remove products of metabolism, and organisms were fed with 10 mg of dry fish food per organism every 3/4 days [43,57]. At the end of the experiment exposure period, organisms were frozen at −20 °C for biochemical analysis.

### 2.3. Quantification of Arsenic

The concentration of Arsenic (As) was determined by inductively coupled plasma-mass spectrometry (ICP-MS) (Agilent 7700x) after acid digestion. In Teflon vessels was added 1.5 mL of HCl and 4.5 mL of HNO_3_ to a 500 mg of homogenized tissue. After 24 h, the Teflon vessels were placed on a heating plate at 115 °C and after 6 h the contents were transferred to a falcon tube. After adding 45 mL of ultrapure water, tubes were centrifuged and then read. Quality Assurance and Quality Control (QA/QC) included the analysis of blanks (vessels with only HCl and HNO_3_), duplicate samples and certified reference material TORT-2 (Lobster Hepatopancreas; 21.6 ± 1.8 mg kg^−1^ As) were in parallel with the samples. Blanks were always below the quantification limit (0.15 μg/g), the mean percentage of recovery for As was 110 ± 4% (*n* = 4) and the coefficient of variation in tissue sample duplicates ranged from 4 to 8%.

### 2.4. Feeding Activity

Furthermore, 21 days after the exposition, the time that the polychaetes needed to feed was recorded. For this assay, commercial fish food was added to the aquariums, and the time needed to polychaetes detect (the time that polychaetes need to come to the sediment surface) and grab the food (time that polychaetes needed to capture the food) in the sediment surface was recorded by video [58]. Nine organisms per condition were used for this endpoint (3 organisms per aquarium × 3 conditions).

### 2.5. Biochemical Analysis

For biochemical analysis, 9 polychaetes (3 per aquaria × 3 replicates) were homogenized in 0.1 M Potassium Phosphate Buffer (pH 7.4) in a sonicator. Homogenates were separated into 3 fractions: one for Lipid Peroxidation assessment; other for Cholinesterase and Electron Transport System, which was centrifuged for 3 min 3300× *g*, at 4 °C, and the rest of the samples were centrifuged for 20 min at 10,000× *g* at 4 °C, for Post-Mitochondrial Fraction (PMS) isolation, used to determine protein and sugars content, superoxide dismutase, catalase and glutathione-S-transferases activity and protein carbonylation content [43]. Fractions were stored at −80 °C until further analysis.

#### 2.5.1. Cholinesterase Activity

Cholinesterase (ChE) activity was determined according to Ellman’s method [59] adapted to microplate [60]. The rate of acetylthiocholine degradation was assessed at 412 nm by measuring the increase in the yellow colour due to the binding of the thiocholine with 5,5-dithio-bis (2-nitrobenzoic acid). The results were expressed as nmol of thiocholine formed per minute per g of Fresh Weight (FW) (ε = 1.36 × 10^4^ M^−1^ cm^−1^), using acetylthiocholine as substrate.

#### 2.5.2. Energy Related Parameters

Total protein content from PMS fraction was determined with Biuret method [61], as performed by [62], using bovine serum albumin (BSA) as standards (0–40 mg/mL). The colourimetric reaction was conducted at room temperature for 10 min and absorbance was measured at 540 nm. Results were expressed in mg per g FW. The Electron Transport System (ETS) activity was measured according to King and Packard [63] methodology with modifications from Coen and Janssen [64]. Absorbance was measured at 490 nm in microplate reader every 25 s for 10 min. The amount of formazan formed was calculated using ε = 15.900 M^−1^ cm^−1^ and the results were expressed in nmol/min per g FW. Sugars were quantified from PMS fraction using the phenol-sulphuric acid method, following the procedure described by Dubois et al. [65]. Absorbance was measured at 462 nm and results were expressed in mg per g FW.

#### 2.5.3. Antioxidant Enzymes

Superoxide dismutase (SOD) activity, measured from PMS fraction, was determined following the method described by Beauchamp and Fridovich [66], with modifications [67]. Absorbance was measured spectrophotometrically in a microplate reader at 560 nm, after 20 min of incubation, at 25 °C. Results were expressed as U per g of FW.

Catalase (CAT) activity was assessed from PMS fraction by the method of Johansson and Borg [68], with modifications described by [62]. Formaldehyde (0–150 μM) standards were used and samples were incubated at room temperature. Absorbance was measured at 540 nm. Results were expressed as U per g of FW.

#### 2.5.4. Biotransformation Enzymes

Glutathione S-Transferases (GSTs) activity was determined based on Habig et al. [69] method with some adaptions [70]. Absorbance was measured spectrophotometrically in a microplate reader at 340 nm (ε = 9.6 × 10^3^ mM^−1^ cm^−1^). Results were expressed in U/g of FW.

#### 2.5.5. Indicators of Oxidative Damage

Lipid Peroxidation (LPO) was measured according to the method of Buege and Aust [71] with modifications [24], by quantifying thiobarbituric acid reactive substances (TBARS), reacting malondialdehyde (MDA) with 2-thiobarbituric acid (TBA). The absorbance was measured at 532 nm. The calculations of the concentration of MDA were made using the molar extinction coefficient (ε = 1.56 × 10^5^ M^−1^ cm^−1^) and expressed in nmol per FW.

Protein carbonylation (PC) levels were analyzed by the quantification of carbonyl groups through the 2,4-Dinitrophenylhydrazine (DNPH) alkaline method described by Mesquita et al. [72] with modifications [73]. Absorbance was read at 450 nm and results were expressed in nmol per FW, using 22,308 M^−1^ cm^−1^ as molar absorptivity of the carbonyl-dinitrophenylhydrazine adduct [72].

### 2.6. Statistical Analysis

Data on As accumulation were submitted to hypothesis testing using permutation multivariate analysis of variance with the PERMANOVA + add-on in PRIMER v6 [74]. The pseudo-F values in the PERMANOVA main tests were evaluated in terms of significance. When the main test revealed statistically significant differences (*p <* 0.05), pairwise comparisons were performed. The t-statistic in the pair-wise comparisons was evaluated in terms of significance among different conditions. The main null hypotheses tested were: (a) considering polychaetes bioaccumulation, for each condition, no significant differences existed among conditions.

For each condition, the parameters Feeding activity, LPO, PC, ETS, Sugars, SOD, CAT, GST, ChE were submitted to hypothesis testing using permutational multivariate analysis of variance, with PERMANOVA+ add-on in PRIMER v6 [74]. Data was analyzed following a one-way hierarchical design, with exposure concentration as the main fixed factor. The null hypothesis tested were: (1) no significant differences existed among As concentrations (0, 0.05 and 0.25 mg/L) for each size (Small, Medium or Large) and for each temperature (16 °C or 21 °C); (2) no significant differences existed between sizes at same concentration of exposure and for each temperature; (3) no significant differences existed between organisms of the same sizes, exposed to the same concentration, among temperatures. The pseudo-F values in the PERMANOVA main tests were evaluated in terms of significance among different concentrations, sizes and temperature. When the main test revealed statistically significant differences (*p <* 0.05), pairwise comparisons were performed. Significance levels (*p <* 0.05) among concentrations and sizes were presented with different letters and significance differences among temperatures were presented with asterisk.

In order to analyse if the global behavioural and biochemical response of *H. diversicolor* was influenced by the size and temperature in the presence and absence of As, the data (fourth root transformed, normalized and the resemblance matrix normalization (Euclidean distance)) were submitted to an ordering analysis performed by Principal Coordinates (PCO), using the PRIMER 6 and PERMANOVA+.

## 3. Results

### 3.1. Accumulation of Arsenic

Total As accumulated in *H*. *diversicolor* tissues from each class size exposed to As at 16 °C and As at 21 °C is represented in Figure 1A,B, respectively. Exposed organisms, independently of the size, showed significantly higher total As concentrations compared to control for both temperatures (Figure 1A,B). An increasing trend in accumulated total As with increasing class size was observed in organisms exposed to 0.05 mg/L of As at 16 °C with significant differences among small and large sizes. On the other hand, a decreasing trend in accumulated total As with increasing class size was observed in organisms exposed to 0.25 mg/L of As at 16 °C with significant differences among small and large sizes.

The same trend was observed in organisms exposed to the same concentration and at 21 °C. No significant differences were observed among small, medium and large size organisms exposed to 0.05 mg/L of As at 21 °C. Small and medium organisms exposed to the highest As concentration at 21 °C presented significantly higher As accumulation than organisms from the same class size exposed to 0.05 mg/L of As. When comparing both temperatures, no significant differences were observed for each class size exposed to different temperatures at the same As concentration.

### 3.2. Feeding Activity

Feeding activity of *H. diversicolor* organisms from each class size is represented in Figure 2, exposed to As and a temperature of 16 °C (Figure 2A) and to As and a temperature of 21 °C (Figure 2B). Large and medium size organisms exposed to As and a temperature of 16 °C needed significantly more time to detect the food compared to the control (Figure 2A). The feeding time was especially long at the highest concentration (0.25 mg/L), being three-fold longer than the control (Figure 2A). No significant changes were observed for small size organisms along with concentration increase. Large size organisms exposed to As and a temperature of 21 °C also significantly increased their feeding time with concentration increase (Figure 2B). Additionally, medium size organisms exposed to As 0.25 mg/L and a temperature of 21 °C have significantly increased time to take food compared to the medium size organisms from the other conditions.

No significant differences were observed for small organisms (Figure 2B). Comparing both temperatures, significant differences were only observed in medium and large size organisms exposed to 0.05 mg/L of As. Medium size organisms took longer to detect food when exposed to a temperature of 16 °C than organisms exposed to the same concentration exposed at a temperature of 21 °C. On the other hand, large organisms exposed to the same As concentration needed more time to detect the food when exposed to a temperature of 21 °C.

### 3.3. Cholinesterase Activity (ChE)

Large size organisms exposed to As 0.05 mg/L and a temperature of 16 °C showed a significant decrease in ChE activity compared to control. Moreover, at the control condition at a temperature of 16 °C, large size organisms have higher ChE activity than small and medium size organisms. Additionally, at the condition of As 0.25 mg/L and a temperature of 16 °C, small organisms have lower ChE activity than large polychaetas, being observed significant differences among sizes (Figure 3A). Regarding exposure to a temperature of 21 °C, significant differences were only detected in small size organisms exposed to the highest As concentration, with organisms presenting significantly lower activity than small size organisms from the remaining conditions (Figure 3B).

Comparing both temperatures, large organisms from all conditions exposed to a temperature of 21 °C presented lower ChE activity than large organisms exposed to a temperature of 16 °C (Figure 3A,B); however, significant differences were only detected in organisms from the control (As 0.0 mg/L). No significant differences were observed among medium organisms between temperatures. Small size organisms not exposed to As (control) presented significantly higher ChE activity at a temperature of 21 °C than organisms from the same class size at a temperature of 16 °C for the same conditions (Figure 3A,B).

### 3.4. Energy-Related Parameters

#### 3.4.1. ETS

Medium size organisms exposed to As 0.05 mg/L at a temperature of 16 °C presented significantly lower ETS levels compared to medium size organisms from the remaining conditions (Figure 4(1A)). Additionally, medium size organisms exposed to As 0.25 mg/L at a temperature of 21 °C have significantly higher ETS levels than same size organisms exposed to As 0.05 mg/L at the same temperature (Figure 4(1B)).

Small size organisms not exposed to As at a temperature of 16 °C and exposed to As 0.25 mg/L at a temperature of 21 °C have significantly lower ETS levels when compared to the other class size organisms from the same condition (Figure 4(1A,B)). Comparing temperature exposure, large size organisms exposed to As showed significantly higher ETS levels when exposed to a temperature of 21 °C (Figure 4(1A,B)).

#### 3.4.2. Sugars

Sugars content showed a decreasing tendency when the organisms from all class sizes were exposed to As at a temperature of 16 °C (Figure 4(2A)). However, when organisms were exposed to As at a temperature of 21 °C, sugars content increased on large size organisms when exposed to 0.25 mg/L As (Figure 4(2B)). Under the same condition, sugars levels were significantly higher on large size organisms and lowest on small size organisms.

Comparing exposure among temperatures, medium class size organisms exposed to a temperature of 21 °C and exposed to 0.25 mg/L of As showed significant higher sugars levels than organisms from the same class size exposed to a temperature of 16 °C. On the other hand, large size organisms exposed to As and a temperature of 21 °C presented significantly higher sugars levels than organisms exposed to a temperature of 16 °C (Figure 4(2A,B)).

#### 3.4.3. Protein Content

Small size organisms exposed to As and at a temperature of 16 °C have significantly higher protein content compared to control (Figure 4(3A)). Protein levels in organisms exposed to a temperature of 16 °C and without As (control) vary significantly among sizes, with small size organisms presenting lower levels and large organisms having higher protein levels. No significant differences were observed for the remaining organisms exposed to a temperature of 16 °C (Figure 4(3A)). For organisms exposed to a temperature of 21 °C, significant differences among class sizes were observed only in small organisms not exposed to As and exposed to 0.25 mg/L of As that had significantly lower protein content than medium and large organisms exposed at the same conditions (Figure 4(3B)). Moreover, all organisms from each class size exposed to a temperature of 16 °C have higher protein content than organisms exposed to a temperature of 21 °C, being significantly different in large organisms from control, medium organisms exposed to 0.05 mg/L and small and medium polychaetes exposed to 0.25 mg/L (Figure 4(3A,B)).

### 3.5. Antioxidant Enzymes

#### 3.5.1. SOD

At 16 °C, small size organisms exposed to 0.05 mg/L As significantly decrease the SOD activity compared to the same size organisms from control. Moreover, large size organisms exposed to the highest As concentration (0.25 mg/L) presented significantly lower SOD activity than polychaetas of the same size from the other conditions (Figure 5(1A)).

Regarding *H*. *diversicolor* exposed to 21 °C, small size organisms exposed to As 0.05 mg/L had significantly lower SOD activity than non-exposed organisms of the same size. On the other hand, large size organisms exposed to As 0.25 mg/L have significantly lower SOD activity than control organisms of the same size (Figure 5(1B)).

Comparing both temperatures, all organisms from all conditions exposed to 21 °C presented higher SOD activity than organisms exposed to 16 °C, with significant differences in small size organisms of all conditions, medium size organisms exposed to 0.05 mg/L and large size organisms exposed to 0.25 mg/L (Figure 5(1A,B)).

#### 3.5.2. CAT

Small size organisms exposed to As significantly increased CAT activity when exposed to 16 °C (Figure 5(2A)), while medium size organisms only significantly increased CAT activity when exposed to As 0.25 mg/L compared to organisms of the same size from remaining conditions. Additionally, large size organisms exposed to As 0.25 mg/L significantly decreased CAT activity compared to the same size of other conditions (Figure 5(2A)). Comparing among conditions, large size organisms exposed to 16 °C without As presented significantly higher CAT activity than small size organisms’. On the other hand, small and medium organisms exposed at As 0.25 mg/L showed higher CAT activity than large ones, being observed with significant differences among sizes.

Regarding organisms exposed to 21 °C, only medium size organisms exposed at 0.05 mg/L of As presented significantly lower CAT activity when compared to organisms from control with the same size (Figure 5(2B)).

Small size organisms exposed to 21 °C presented higher CAT activity than organisms exposed to 16 °C for all conditions. Medium organisms presented significantly higher CAT levels when exposed to 21 °C without As and at the highest As condition, than medium organisms exposed to the same concentrations at 16 °C. Large polychaetes only presented significantly higher CAT levels when exposed to 21 °C at 0.25 mg/L As. (Figure 5(2A,B)).

### 3.6. Biotransformation Enzymes (GSTs)

Small size organisms exposed to As 0.25 mg/L and 16 °C showed significantly higher GSTs activity than small size control organisms (Figure 6A). Medium size organisms exposed to As 0.25 mg/L and 16 °C have significantly higher CAT activity than medium size organisms exposed to control and to As 0.05 mg/L. Furthermore, large size organisms of As 0.25 mg/L and 16 °C showed significantly higher activity than the others of the same size conditions. Additionally, in control condition, medium size organisms have significantly higher activity than organisms from the remaining sizes. Nevertheless, small size organisms exposed to As 0.25 mg/L at 16 °C have significantly lower activity from the condition (Figure 6A).

Small size organisms exposed to As 0.25 mg/L at 21 °C presented significantly lower activity than small size of As 0.05 mg/L. Additionally, large size organisms from control at 21 °C presented significantly higher activity than small and medium size organisms (Figure 6B).

Comparing both temperatures, almost all organisms from all conditions exposed at 21 °C presented significantly higher GSTs activity than organisms exposed at 16 °C, except medium size organisms not exposed to As (Figure 6A,B).

### 3.7. Indicators of Oxidative Damage

#### 3.7.1. LPO

No significant differences in lipid peroxidation levels were observed on organisms exposed at 16 °C (Figure 7(1A)), independently of the class size or the exposure to As.

Lipid peroxidation levels significantly increased for large size organisms exposed to the highest As concentration and at 21 °C (Figure 7(1B)). Moreover, large organisms exposed at 21 °C for all As conditions and medium size organisms exposed to As 0.05 mg/L at 21 °C have significantly higher lipid peroxidation levels compared to *H*. *diversicolor* organisms from the same class sizes exposed at 16 °C (Figure 7(1A,B)).

#### 3.7.2. PC

Protein carbonylation levels significantly increased in large organisms compared to control conditions in large organisms exposed to the highest As concentration and at 16 °C (Figure 7(2A)). Regarding exposure to 21 °C, small size organisms exposed to As presented significantly higher protein carbonylation than organisms not exposed to As (Figure 7(2B)). Moreover, small organisms exposed at 21 °C without As (0.0 mg/L) presented significantly lower protein carbonylation when compared to the same class size exposed at 16 °C and not exposed to As (Figure 7(2A,B)).

### 3.8. Multivariate Analysis

Principal coordinates analysis (PCO) graph obtained for *H. diversicolor* from different class sizes exposed to As at 16 °C and 21 °C. As accumulation, feeding activity and biochemical parameters are shown in Figure 8. The PCO axis 1 accounted for 32% of total data variation, separating medium and large organisms exposed at 21 °C small size organisms to As 0.05 mg/L at 21 °C, and large organisms on control and medium organisms to As 0.25 mg/L at 16 °C in the positive side and the majority of organisms exposed at 16 °C and small size organisms at the control and As 0.25 mg/L at 21 °C in the negative side (Figure 8). Organisms separated on the positive side are associated with LPO, GSTs and ETS, levels (r > 0.8). PCO axis 2 explained 28.9% of total data variation, with almost all organisms exposed at 16 °C, except small size organisms from control, and large size organisms exposed to As 0.05 mg/L and 0.25 mg/L at 21 °C on the positive side, being associated to the parameters PROT, PC and FA (r > 0.8). On the negative side were separated small and medium size organisms and large size control organisms exposed at 21 °C and small size organisms from control at 16 °C, which are associated with the SOD activity (r > 0.8) (Figure 8).

## 4. Discussion

Polychaetes are being widely used as bioindicators in ecotoxicological assays to evaluate behavioural and biochemical alterations caused by contaminants, such as metalloids and climate change [30,75,76,77]. The species *H*. *diversicolor* is one of the most used species, usually showing high sensitivity to low levels of these alterations, being translated in alterations of behaviour, energy status, oxidative stress, oxidative damage and cholinesterase inhibition [30,43]. Studies usually use adult organisms, not existing information about how contaminants and climate alterations impact organisms from different ages. Thus, this study aimed to investigate how contamination by As impacts the polychaete *H. diversicolor* from different sizes (small, medium and large organisms) exposed at two different temperatures (16 °C and 21 °C).

Total As accumulation in polychaetes tissues showed an accumulation pattern related to As concentration. Small organisms presented significantly higher accumulation levels in the highest As concentration of exposure, compared to large organisms, being observed a decreasing trend in accumulated total As with increasing class size in organisms exposed to 0.25 mg/L. Previous studies demonstrated that elements concentration in polychaetes tissues might also be influenced by their weight. In fact, Garcês and Costa [78] found higher concentrations of the metals Zn, Cu and Cd in the small wet-weight classes of the polychaete *Marphysa sanguinea* in areas with higher levels of these metals. The observed decline of elements in the bigger worms could be a result of growth-dilution or different metabolic routes involved in metal accumulation and excretion [78]. However, comparing As accumulation among studied temperatures, in addition to the increasing trend in As accumulation in small and medium organisms exposed at 21 °C, no significant differences were observed. However, a previous study indicated that As concentrations in *Mytillus galloprovinciallis* were significantly higher after 28 days of exposure to As (1.0 mg/L) at 21 °C than at 17 °C [79].

*H*. *diversicolor* are important organisms in the ecosystems they inhabit since their functional traits, such as burrowing and feeding behaviour, provide sediment irrigation, oxygenation and particle mixing [49,52] and alterations on these bioturbation activities can provide ecological consequences [80]. In this study, it was detected that exposure to As induced negative effects on medium and large organisms’ behaviour, since exposed organisms needed more time to detect and grab the food with increasing concentration, independently of temperature. These behavioural changes (longer time to feed) may have intrinsic ecological effects since slower organisms are more susceptible to predators. Moreover, sediment oxygenation may be reduced, affecting organisms that rely on bioturbation. Additionally, Urban-Maling et al. [81] demonstrated that *H*. *diversicolor* exposed to graphene nanoflakes resided deeper in the sediment than controls, suggesting an escape response to graphene. Thus, this parameter had not been measured in our study. If organisms are deeper in the sediment, it could explain the increased time that polychaetes needed to detect food. Moreover, the reduction in feeding activity will conduct in a decrease in sediment oxygenation, which is essential for maintaining infauna diversity [52]. Some studies demonstrated that temperature increases have an important role in organisms’ activity, such as feeding rate or swimming performance [82,83]. In our study, significant differences among temperatures were detected only in medium and large size organisms exposed to 0.05 mg/L of As, with medium size organisms needing less time to detect food when exposed at 21 °C than organisms exposed at the same concentration exposed at 16 °C. On the other hand, large organisms exposed to the same As concentration needed more time to detect the food when exposed at 21 °C. A study conducted with *Dicentrarchus labrax* juveniles demonstrates that these species have their swimming performance improved when the water increased to a maximum temperature of 25 °C [82]. Additionally, an increase in feeding rate with temperature increase was observed for *Amphiprion clarkia* post-larval stage [83]. On the other hand, juveniles of *Salmo trutta* L. demonstrated that the temperature increase receded their food intake, activity and swimming endurance when exposed to temperatures higher than 18 °C [84].

One of the main known functions of ChEs is to catalyze the hydrolysis of the neurotransmitter acetylcholine that returns an activated cholinergic neuron to its resting state [85,86] as being vital for normal muscular function and behaviour of most species [87]. Inhibition of ChE is well-recognized as a biomarker of exposure to several contaminants, including metals and As [88,89]. Several studies observed that ChE activity is inhibited readily by arsenite, and Page and Wilson [90] stated that when arsenite bind to acetylcholinesterase, it forms a diester with two tyrosines. The formation of these covalent bonds explains the slow reaction of the enzyme with arsenite and the slow dissociation of arsenite [90]. In this work, the activity of the enzyme ChE, an enzyme responsible for normal muscular and behavioural functions [85,86,87], seems to not compromise small and medium organisms exposed at 16 °C. However, large organisms significantly diminished their ChE activity when exposed to arsenic, which may indicate that the mobility of this organism could be compromised, and thus, needing more time to leave the sediments for feeding, and reveal the ability of As to inhibit ChE activity. Though, at 21 °C, only small size organisms have ChE activity compromised when exposed to As 0.25 mg/L at 21 °C. Previous studies demonstrated that *H. diversicolor* have their ChE activity reduced when exposed to As [23] and other contaminants, such as polystyrene nanoplastics [43] and inorganic mercury [76]. This pattern of negative effects of ChE activity was also observed on *Diopatra neapolitana* when exposed to multi-walled carbon nanotubes [91], *Donax trunculus* exposed to microplastics [92] and *Tegillarca granosa* exposed to bisphenol A and microplastics [93]. On the other hand, the bivalves *M. galloprovincialis* and *Corbicula fluminea* have their ChE activity increased proportionally with temperature increase, achieving the maximum activity at 38 °C and 45 °C [94], indicating that temperature increase may contributes to the increase in ChE activity, as observed in small organisms in our study.

Energy metabolism has an important function in organisms’ survival and function, and in stress tolerance and adaptation [95]. Moreover, under environmental disturbances, organisms can increase energy expenditure, being considered a mechanism of cellular protection [96]. The parameter ETS activity has been used as a measure of metabolic capacity on invertebrates in order to response to environmental disturbances, as chemical and metal stress [22,67,75,97]. In fact, in this study, the obtained results suggested that temperature increase contributed to an increase in *H. diversicolor* metabolism in large size organisms exposed to As. Some previous studies demonstrated that this species have their ETS activity increased when exposed to climate alterations, such as a pH decrease [98] or when exposed to others contaminants, such as polystyrene nanoplastics [43] and multi-walled carbon nanotubes [91]. On the other hand, in stressful situations, to avoid energy expenditure which may prevent them from greater damages, a decrease in ETS activity may occur, as observed in medium organisms exposed to As 0.05 mg/L at 16 °C from our study and as detected in *H. diversicolor* organisms exposed to 5 μg/L of mercury and mercury and acidification [75] and in *M. galloprovincialis* exposed to thermal stress and Arsenic [79].

Sugars and protein (PROT) act as major sources of energy in polychaetes in stressful environments [99]. In this study, organisms when exposed to As at 16 °C, use sugar as the main source of energy, lowering their sugar reserves. Previous studies also showed a decrease in sugar content when exposed to mercury and acidification [75] and pharmaceuticals [67], indicating that at these conditions, polychaetes used sugars to obtain energy. On other hand, large size organisms increased their sugar content when exposed to As 0.25 mg/L at 21 °C, indicating that this species, under stressful conditions, as As contamination and temperature increase, may prevent energy expenditure in specific processes (e.g., limiting their use for polychaetes burrowing activity) or to fuel up defence mechanisms were using other energy sources, such as lipids.

Regarding PROT content, it was observed a pattern of protein content among temperatures, with higher protein levels at 16 °C than at 21 °C. Moreover, small size organisms at 16 °C have PROT content increased when exposed to As. Other studies also revealed that energy reserves of *H*. *diversicolor* exposed to mercury and/or seawater acidification diminished [75,98], demonstrating that *H*. *diversicolor* polychaetes tried to prevent oxidative stress by increasing organisms’ antioxidant defences (namely increasing SOD and CAT activity), leading to energy expenditure. On the other hand, a suppression of protein synthesis was also previously observed as a marker associated to environmental stress and an hallmark of metabolic rate depression [95].

Previous studies showed that contamination and climate change induced several harmful biochemical effects, such as oxidative stress in exposed organisms [30,77], which resulted from the overproduction and accumulation of reactive oxygen species (ROS). To prevent these injuries, organisms are able to activate their defences, such as antioxidant (SOD and CAT) and biotransformation (GSTs) enzymes to eliminate ROS and toxic compounds formed from the metabolism of oxidized molecules, such as lipid hydroperoxides, avoiding cell and protein damage [100]. In this study, it was detected that increased antioxidant activity of the enzymes SOD and CAT when *H*. *diversicolor* was exposed at 21 °C. In addition, small and large size organisms have their superoxide dismutase (SOD) activity lower when exposed to arsenic, indicating that the activity of this enzyme was inhibited. However, previous studies showed that usually SOD activity of *H*. *diversicolor* usually increased when exposed to contaminants, such as mercury [75]. The same pattern was observed with *Diopatra neapolitana* when exposed to arsenic and to metals and As [22,24] and with the bivalve *Ruditapes philippinarum* exposed to As [101].

Regarding catalase (CAT) activity, in small and medium organisms exposed at 16 °C, the activity of this enzyme increased in As exposed organisms, but it was inhibited in large size organisms. Previous studies also showed an activity reduction in this enzyme when exposed to contaminants, such as polystyrene nanoplastics, indicating an adaptative mechanism as a possible explanation for this inhibition [43]. However, CAT inhibition will cause inefficiency of defence, contributing for the increasing of LPO levels, as observed when *H*. *diversicolor* was exposed to mercury [75] and when *D. neapolitana* and *R. philippinarum* were exposed to arsenic [22,101].

GSTs are responsible for detoxifying the xenobiotics and metabolites produced from oxidative stress, as lipid peroxides by-products and are widely used to evaluate the detoxification capacity of organisms. Previous studies demonstrated that As induces GSTs activity [22,102,103]. In fact, in this work, *H. diversicolor* increased GSTs activity when exposed to As 0.25 mg/L at 16 °C. Previous studies also reported that *H*. *diversicolor* increased GSTs activity when exposed to As [23] and also to other elements, such as soluble Ag [104] and mercury confirming their detoxification capacity [75]. The highest GSTs activity in medium and large organisms, when exposed to 0.25 mg/L, may had contributed for the lower As concentrations were observed in polychaetes of this class size exposed at this concentration. In our study, organisms exposed at 21 °C presented significantly higher GSTs activity than organisms exposed at 16 °C for almost all class sizes and As conditions. A study with *M. galloprovincialis* with similar conditions showed that the combination of As and 21 °C stimulates GSTs activity [79]. This increase in GSTs activity, also occurs when *Tigriopus japonicus* is exposed to 35 °C for 60–120 min due to ROS increase [105].

In this study, it was observed that the combination of As and temperature rise (21 °C), contributed to the increased of lipid peroxidation (LPO). However, no cellular damage was observed in all size organisms exposed to normal temperature condition (16 °C) with the same As concentration (0.05 and 0.25 mg/L). Alves et al. [106] demonstrated that when the antioxidant mechanism fails to manage the stress, the LPO increase causes cellular damage, suggesting that antioxidant enzymes, such as SOD and CAT were not able to eliminate ROS. In fact, SOD activity was inhibited in large size organisms from As 0.25 mg/L at 21 °C, being not able to eliminate ROS and conducing to cellular damage despite higher CAT activity. Furthermore, some studies also demonstrated that the polychaete *D*. *neapolitana* always presented higher LPO levels when exposed to stress [22,107].

Regarding protein carbonylation (PC), in this study it was observed that large organisms from As 0.25 mg/L at 16 °C and small organisms when exposed to As at 21 °C have increased PC levels, suggesting potential protein damage. PC is one of the most harmful and irreversible oxidative protein modifications. Carbonyl stress is related to biomolecules malfunction, immunogenicity, inflammation, cell toxicity and apoptosis [108]. Thus, under stressful conditions, such as exposure to contaminants, is expected to observe an increase in PC levels in exposed organisms, as observed in *H*. *diversicolor* exposed to As and polystyrene nanoplastics [23]. *D. neapolitana* worms from higher trace elements contaminated areas at Ria de Aveiro, Western Portugal also presented higher PC levels were also observed in Giménez et al. [109].

## 5. Conclusions

The present study demonstrated that *H*. *diversicolor* from different sizes behave differently among As concentrations and temperatures. These results are helpful to understand the effects of contaminants and climate change at different ages, and how it could affect the *H*. *diversicolor* population. Moreover, in this work, it was possible to observe that, for some parameters (as ETS, SOD, GSTs and LPO) temperature increases combined with As have a higher impact on *H*. *diversicolor* at almost every different size.

Our findings show that in terms of behaviour and oxidative stress, large size organisms, in general, are the most affected size class among the observed ages since they need more time to detect the food, and have higher cellular damage. Additionally, medium size organisms seem to be the most well-adapted group class, since this group of organisms presented fewer alterations in the tested parameters. Nevertheless, this study showed that the combination of As and temperature revealed an additive effect since the metabolic capacity and the activity of antioxidant defences increased in comparison to As acting alone, mainly in large size organisms.

Furthermore, this study highlighted the importance of integrating behavioural and biochemical responses as an approach to understanding the mechanistic bases of stress responses and interpreting their ecological consequences. Changes in behaviour may suggest possible consequences for the *H. diversicolor* population since slower polychaetes may be more susceptible to predators. Additionally, the functions of the ecosystem may also be altered due to the decrease in burrowing activity, since organisms may also not promote proper sediment oxygenation.

Additionally, the present study also showed that there is a need to understand how these environmental alterations will impact organisms in different life stages and to check a possibly compromised generation of worms. Since this species has an important role in the environment, any alterations can unbalance the ecosystem that this species inhabits.

## Figures and Tables

**Figure 1 ijerph-19-15426-f001:**
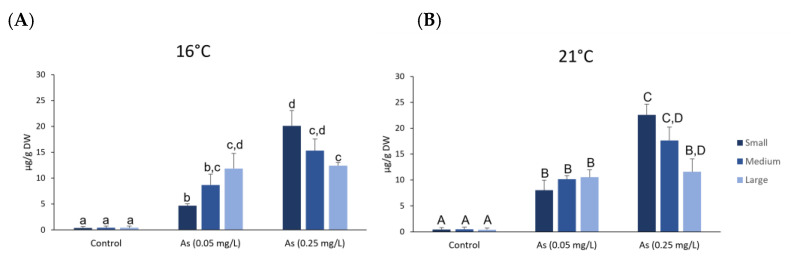
Total Arsenic in *Hediste diversicolor* tissues from different class sizes after exposure to Arsenic at 16 °C (**A**) and at 21 °C (**B**). Different letters represent significant differences (*p <* 0.05) between conditions (lowercase letters for 16 °C; uppercase letters for 21 °C).

**Figure 2 ijerph-19-15426-f002:**
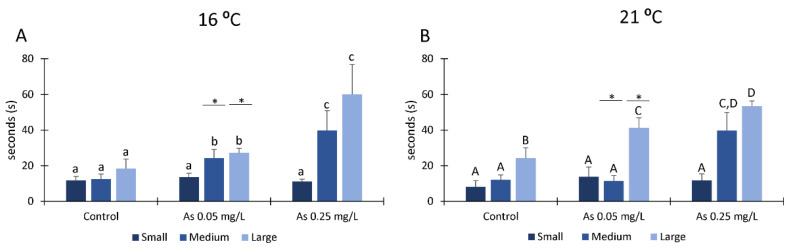
Feeding activity (time needed to detect and grab the food) 21 days after exposure to Arsenic and a temperature of 16 °C (**A**) and Arsenic and a temperature of 21 °C (**B**) of *Hediste diversicolor* from different class sizes. Different letters represent significant differences (*p <* 0.05) between conditions (lowercase letters for 16 °C; uppercase letters for 21 °C). Asterisk (*) represents significant differences (*p <* 0.05) among temperatures for the same class size.

**Figure 3 ijerph-19-15426-f003:**
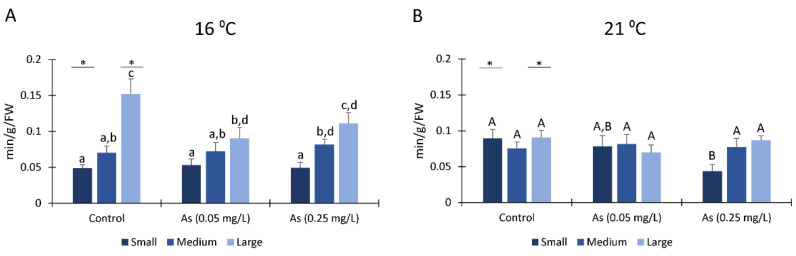
Cholinesterase (ChE) activity measured in *Hediste diversicolor* from different class sizes after 28 days of exposure to Arsenic at a temperature of 16 °C (**A**) and 16 °C (**B**). Different letters represent significant differences (*p* ≤ 0.05) between conditions (lowercase letters for 16 °C; uppercase letters for 21 °C). Asterisk (*) represents significant differences (*p <* 0.05) among temperatures for the same class size.

**Figure 4 ijerph-19-15426-f004:**
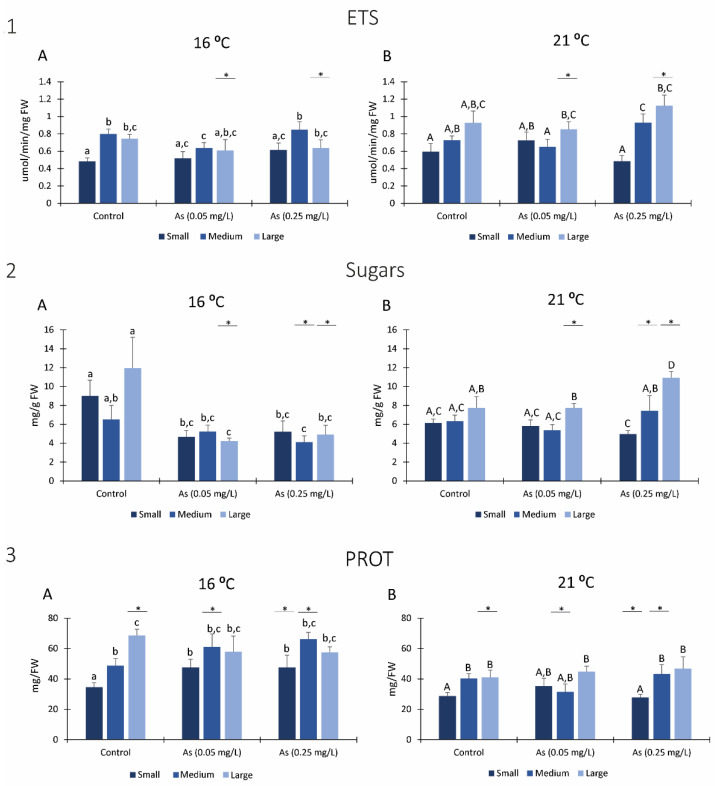
Metabolism related parameters. (**1**) Electron transport system (ETS) at a temperature of 16 °C (**A**) and 21 °C (**B**) of *Hediste diversicolor* after 28 days of exposure to Arsenic. (**2**) Sugars content measured in *H*. *diversicolor* from different class sizes after 28 days of exposure to Arsenic at a temperature of 16 °C (**A**) and 21 °C (**B**). (**3**) Protein content 28 days after exposure to Arsenic and a temperature of 16 °C (**A**) and Arsenic and a temperature of 21 °C (**B**) of *Hediste diversicolor* from different class sizes. Different letters represent significant differences (*p <* 0.05) between conditions (lowercase letters for 16 °C; uppercase letters for 21 °C). Asterisk (*) represents significant differences (*p <* 0.05) among temperatures for the same class size.

**Figure 5 ijerph-19-15426-f005:**
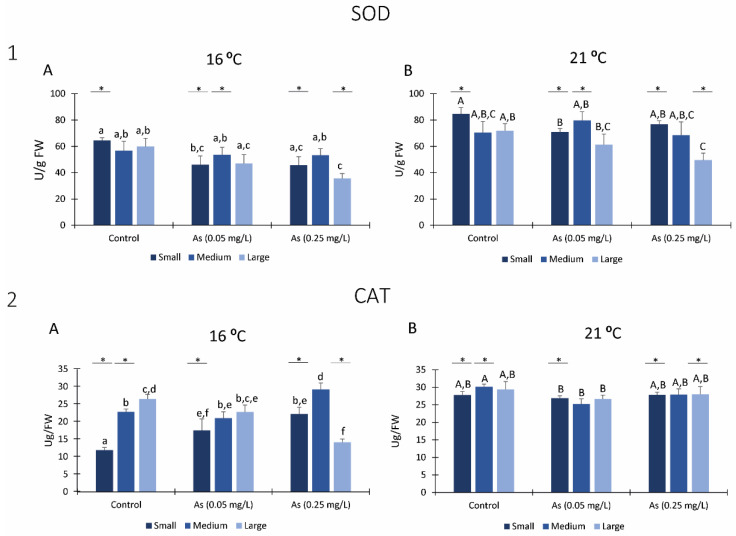
Antioxidant enzymes: (**1**) Superoxide dismutase activity (SOD) measured in *H*. *diversicolor* from different class sizes after 28 days of exposure to Arsenic at 16 °C (**A**) and 21 °C (**B**). (**2**) Catalase (CAT) activity measured in *H*. *diversicolor* from different class sizes after 28 days of exposure to Arsenic at 16 °C (**A**) and 21 °C (**B**). Different letters represent significant differences (*p <* 0.05) between conditions (lowercase letters for 16 °C; uppercase letters for 21 °C). Asterisk (*) represents significant differences (*p <* 0.05) among temperatures for the same class size.

**Figure 6 ijerph-19-15426-f006:**
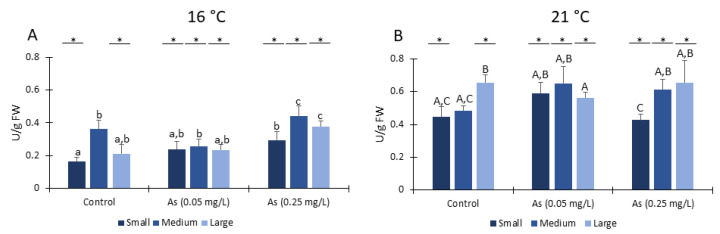
Glutathione-S-transferases (GSTs) activity measured in *H*. *diversicolor* from different class sizes after 28 days of exposure to Arsenic at 16 °C (**A**) and 21 °C (**B**) of *Hediste diversicolor* after 28 days of exposure. Different letters represent significant differences (*p <* 0.05) between conditions (lowercase letters for 16 °C; uppercase letters for 21 °C). Asterisk (*) represents significant differences (*p <* 0.05) among temperatures for the same class size.

**Figure 7 ijerph-19-15426-f007:**
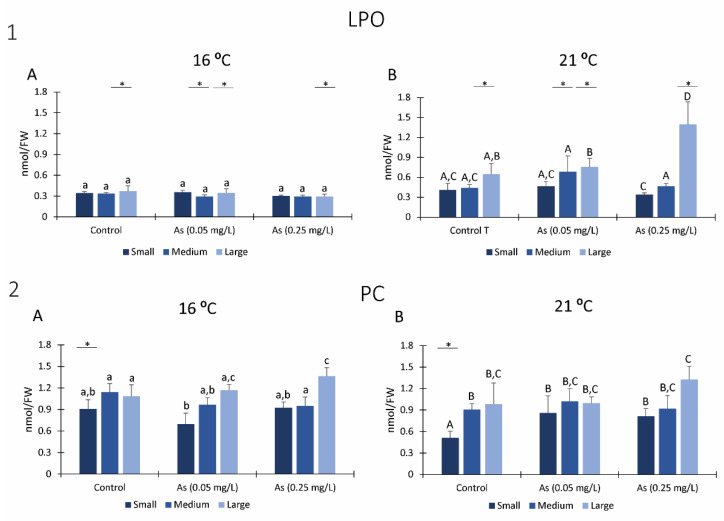
Indicators of oxidative damage: (**1**) Lipid peroxidation (LPO) measured in *Hediste diversicolor* from different class sizes after 28 days of exposure to Arsenic and at 16 °C (**A**) and 21 °C (**B**). (**2**) Protein carbonylation (PC) measured in *Hediste diversicolor* from different class sizes after 28 days of exposure to Arsenic and 16 °C (**A**) and 21 °C (**B**). Different letters represent significant differences (*p <* 0.05) between conditions (lowercase letters for 16 °C; uppercase letters for 21 °C). Asterisk (*) represents significant differences (*p <* 0.05) among temperatures for the same class size.

**Figure 8 ijerph-19-15426-f008:**
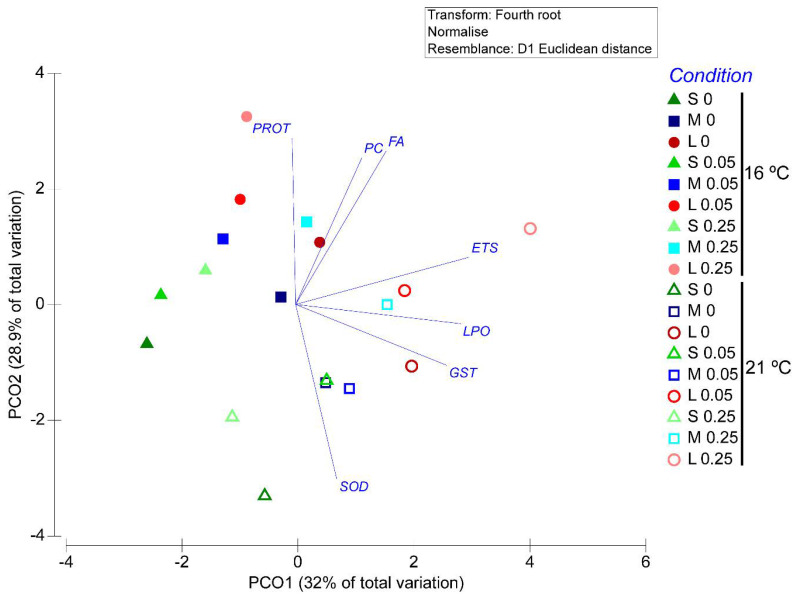
Centroids ordination diagram (PCO) based on feeding activity and biochemical parameters, measured in *Hediste diversicolor* exposed to different As concentrations and sizes (0; 0.05; 0.25 mg/L and Small (S), Medium (M) and Large (L)) from both temperatures (16 °C and 21 °C). Pearson correlation vectors are superimposed as supplementary variables, namely feeding activity and biochemical data (r > 0.8).

## Data Availability

Not applicable.

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
