# Peer review of "Biochemical and Behavioural Alterations Induced by Arsenic and Temperature in Hediste diversicolor of Different Growth Stages"

_ijerph, 2022, doi:10.3390/ijerph192315426_

Round 1

Reviewer 1 Report

In this environmental toxicology study, the authors examine the effects of two different temperatures and two different concentrations of As (plus control) on the benthic polychaete marine worm Hediste diversicolor at three different sizes (ages). The observations are grouped into As accumulation, feeding activity, "neuro-transmission" (cholinesterase activity), metabolic parameters (three), antioxidant enzymes (two), "biotransformation enzymes" (glutathione-S-transferase), and oxidative damage markers (two). 

Overall, it is certainly important to understand the effects of pollution, and to take into account the likely effects of global warming.  Thus, this seems like a potentially useful contribution.

I come to this manuscript as a novice, so I am unfamiliar with the standards and practices or the field, and my comments should be interpreted within that context.

1. Nowhere is it stated what chemical form the As is in. Surely the effects of the element depends on the chemical form in which it is presented. For the water changes, I assume that the new water was brought to the appropriate temperature before changing, but this was not stated.

2. The numbers of animals for the different size sample varies from 5 for the largest size, to 10 for intermediate and 20 for the smallest size. A priori, it seems like this might affect the statistical analysis. The data are presented as bar graphs with error bars. I would prefer to see each replicate represented directly (scatter plot), so the reader can better gauge the reproducibility of the data. Also, I was unable to find an adequate explanation of the letters that are apparently used to indicate statistical significance within each temperature sample.

3. I appreciate the depth of sampling involved in this work, something like 11 different measurements for each of the six temperature and As exposures.  Nonetheless, I find the interpretations of these data to be overly broad and involving problematic assumptions.  For example, equating cholinesterase activity from a whole animal homogenate with neurotransmission is unwarranted: 1) there are many neurotransmitters besides acetylcholine: 2) even at the presumably cholinergic excitatory neuromuscular junctions, neurotransmission entails many steps besides the hydrolysis of acetylcholine; 3) cholinesterase activity is not localized to cholinergic synapses.  Finally, there is no reason to assume that all aspects of neurotransmission would be affected to a similar degree or in the same direction by these treatments.  Similarly, the biomolecular content of the animals (here, sugar and protein content) reflects the balance between synthesis and degradation of the material in question. Thus, I don't think it is correct to assume that lower content reflects higher utilization...couldn't it also reflect lower rates of synthesis?

4.  I had a difficult time trying to understand what conclusions might be drawn from this work.  Perhaps it would be useful to start by identifying which processes if any vary significantly with body size/age, which vary significantly with changes in temperature, and which ones vary significantly with As exposure, and then look for synergistic interactions. This is complicated by not being sure about which of the many measurements are revealing significant differences (see item 2 above).

5. I am deeply sympathetic to the challenges of writing in a second language, which I assume is the case.  The English is adequate for the most part, but in several places (too numerous for me to list here) the authors' meaning is unclear or ambiguous.

Author Response

The authors would like to thank the reviewers for all their comments and suggestions that greatly improved the quality and clarity of the manuscript. All comments were taken into consideration and the manuscript was revised accordingly. All changes are highlighted in blue.

General Comments

Reviewer 1

General comment:

In this environmental toxicology study, the authors examine the effects of two different temperatures and two different concentrations of As (plus control) on the benthic polychaete marine worm Hediste diversicolor at three different sizes (ages). The observations are grouped into As accumulation, feeding activity, "neuro-transmission" (cholinesterase activity), metabolic parameters (three), antioxidant enzymes (two), "biotransformation enzymes" (glutathione-S-transferase), and oxidative damage markers (two). 

 Overall, it is certainly important to understand the effects of pollution, and to take into account the likely effects of global warming.  Thus, this seems like a potentially useful contribution.

 I come to this manuscript as a novice, so I am unfamiliar with the standards and practices or the field, and my comments should be interpreted within that context.

Response to General Comment: The authors like to express their gratitude for the reviewer' insightful remarks and ideas. All of these were considered in the revised version of the manuscript.

  1. Nowhere is it stated what chemical form the As is in. Surely the effects of the element depends on the chemical form in which it is presented. For the water changes, I assume that the new water was brought to the appropriate temperature before changing, but this was not stated.

R: The information about the chemical form the As was included in the manuscript (lines 148-150). Additionally, for the water changes, the new water was in the appropriate temperature (line 162). This information was added to the manuscript.

  1. The numbers of animals for the different size sample varies from 5 for the largest size, to 10 for intermediate and 20 for the smallest size. A priori, it seems like this might affect the statistical analysis. The data are presented as bar graphs with error bars. I would prefer to see each replicate represented directly (scatter plot), so the reader can better gauge the reproducibility of the data. Also, I was unable to find an adequate explanation of the letters that are apparently used to indicate statistical significance within each temperature sample.

R: Thank you for your comment. In fact, for each analysis, it was used the same number of organisms (3 polychaetes per replicate x 3 replicates per condition = 9 polychaetes in total per condition). However, for experiments, we used a higher number of juveniles since we assumed that they should be more sensible and some mortality was expected to occur.  Nevertheless, no mortality was detected, but for all analyses, the same number of organisms was used. This was explained in the manuscript (lines).  

Regarding data presentation, although the authors agree that representing the data in a scatter plot is more informative, we decided to keep the bar graphs with error bars, since changing all the figures would be hard and time-consuming work. But if the reviewer considers that this change is really necessary to publish the paper, the authors will try to make an effort to change the images.

Regarding the letters, different letters represent significant differences (p≤0.05) among all conditions (when no differences were found, the same letter was used, if there are differences, it was used a different letter). It was used lowercase letters for 16 °C and uppercase letters for 21 °C.

  1. I appreciate the depth of sampling involved in this work, something like 11 different measurements for each of the six temperature and As exposures.  Nonetheless, I find the interpretations of these data to be overly broad and involving problematic assumptions.  For example, equating cholinesterase activity from a whole animal homogenate with neurotransmission is unwarranted: 1) there are many neurotransmitters besides acetylcholine: 2) even at the presumably cholinergic excitatory neuromuscular junctions, neurotransmission entails many steps besides the hydrolysis of acetylcholine; 3) cholinesterase activity is not localized to cholinergic synapses.  Finally, there is no reason to assume that all aspects of neurotransmission would be affected to a similar degree or in the same direction by these treatments.  Similarly, the biomolecular content of the animals (here, sugar and protein content) reflects the balance between synthesis and degradation of the material in question. Thus, I don't think it is correct to assume that lower content reflects higher utilization...couldn't it also reflect lower rates of synthesis?

R: Thank you for your comment. The data interpretation regarding neurotransmissions was revised. Regarding the sugar and protein content, previous studies indicated that high energy expenditure (sugars, lipids and proteins) during exposure to stressful conditions were commonly observed  [1,2]. However, suppression of protein synthesis was also previously observed as a marker associated with environmental stress and a hallmark of metabolic rate depression. So, the authors agreed that lower protein levels may also reflect lower rates of synthesis. This was modified in the manuscript (lines 603-605).

  1. I had a difficult time trying to understand what conclusions might be drawn from this work.  Perhaps it would be useful to start by identifying which processes if any vary significantly with body size/age, which vary significantly with changes in temperature, and which ones vary significantly with As exposure, and then look for synergistic interactions. This is complicated by not being sure about which of the many measurements are revealing significant differences (see item 2 above).

R:  The conclusion section was rewritten.

  1. I am deeply sympathetic to the challenges of writing in a second language, which I assume is the case.  The English is adequate for the most part, but in several places (too numerous for me to list here) the authors' meaning is unclear or ambiguous.

R: The English was carefully revised.

Reviewer 2 Report

The manuscript titled " Biochemical and behavioural alterations induced by Arsenic 2 and temperature in Hediste diversicolor of different growth stages " determined the impact of As combined with the temperature changes on Hediste diversicolor analyzing several biochemical and behavioral bookmarkers. The authors found oxidative damage on lipids and proteins as well as a synergism in the combination of 21oc and As exposure for small and lager invertebrates. This topic is pertinent given the need to assess the risk of metals such as As in a changing environment, namely warning changes. However, I found that several aspects required more development or key information. In addition to that, there are serious issues related to missing methodological information (QC/QA controls) that should be addressed. Considering these changes, I would recommend this article for publication in this journal.

General comments:

1-The authors conducted a series of experiments including As exposure as well as temperature changes (16oC versus 21oC). Do you think the way your tested warming climate (from 16 to 21oC as described in lines 154-155) is a pertinent and realistic way to address this point? A temperature increasing daily 1oC is a realistic way to address global warming?

2- Methodology section required further information to better evaluate the experiments performed:

-Exposure experiments:

Do you measure As and other metals (Hg, Se) in food used during the nursery and exposure? Did As concentrations were stable during your experiments? Any stability test performed to check that? Did you make any comparison between nominal As concentrations versus As concentration in water? What about the sediment used? Did you check there was not any potential As contamination from the sediments? Did you perform any desorption and depuration analyses before measuring As in H. diversicolor?

Metal measurements:

It is important to show your QC/QA control results. In section 2.3, you wrote that you used them but nothing is indicated later about these results. Any As contamination in blanks? What kind (source, biological sample) of certified reference material you used? How many replicates (n)? What is the detection limit for As? What does “bias error of the chemical analysis was less than 10% » mean?

Biomarkers:

In the feeding activity measurement, it is not clear how the author estimated the time needed for your animals to detect and catch food? Any video recording analyses? How to distinguish detection and capture? References about this test should be added. Please provide more details about this endpoint since you obtained as results several seconds. How to ensure a good quality of such estimations?

In section 2.5, it is necessary to better explain the different steps followed to obtain the samples for each biomarker. Even try to do a methodological scheme for each fraction or step. I think some improvements are needed and more importantly, to stay coherent with what you wrote in lines 189-207. In what fraction did you measure SOD and CAT? In the PMS fraction, you quantified proteins, sugars (as described in section 2.5.2) but that is not included at the beginning of section 2.5. FW indicates wet weight?

3- In this work, you generated sufficient data to get insights about the impact of As + temperature based on a combination of biochemical and behavioral endpoints, but you showed and discussed such results individually (biomarker by biomarker) without doing any global consideration, link, or correlation between these measurements. Why not prepare a PCA figure as an exploratory approach to distinguish similarities or differences in your results? To explore the link between the endpoints should be necessary. I was expecting something like that at the end of the manuscript.

Minors comments.

-Title: Is it necessary to have a capital latter at the beginning of the word Arsenic?

-Abstract:

I really enjoyed reading the last sentences, but more development of this idea should be added in the discussion section.

-Keywords - the word “arsenic” is already in the title.

-Introduction

Introduction section presents key information, but I was wondering if it is really needed in the first paragraph. Start your introduction talking about As, it is suitable to mention nothing about organic contamination (you did not address this point in your document).

Line 72: Are you saying as can show biomagnification? If yes, in what condition? What type of ecosystems?

 -Materials and Methods

Line 150: What was the stability of as in your experiments? Do you have a ratio of nominal As concentrations versus As measured in water?

Line 157: Please show some metal concentrations already found in the area. Just to confirm you are working with environmentally realistic As concentrations.

Line 181: At what temperature the PMS fractions were kept?

-Results:

In fig.1 you are showing metal concentrations in ug of As per weight of your organisms (g,  FW). Did you have any idea about the weight variation in your organisms during the 21 days of metal exposure? That should be interesting for results obtained for As exposure at 0,25 mg/L where As concentration decreased. Please, indicate the statistical test used in your comparisons. N=3 is a limited replicate number for some tests.

In fig. 2:  it is not common that different life stages differed in their food capacity, even in their food type to ingest. So, why to make comparison between different class sizes? It should be more important to do that among As and T conditions than among class size.

Discussion:

-Why not to discuss the results obtained in figure 1 for th4 three class sizes exposed to 0.25 mg/L?

-The results regarding cholinesterase (ChE) activity required more development. How metal can affect the activity of this biomarker? More references to this point are expected to be added.

-At times, the comparison of the results obtained is limited to organic contaminants, so why not to make more efforts in making these comparisons with other metals or elements?

Author Response

The authors would like to thank the reviewers for all their comments and suggestions that greatly improved the quality and clarity of the manuscript. All comments were taken into consideration and the manuscript was revised accordingly. All changes are highlighted in blue.

General Comment: The manuscript titled " Biochemical and behavioural alterations induced by Arsenic 2 and temperature in Hediste diversicolor of different growth stages " determined the impact of As combined with the temperature changes on Hediste diversicolor analyzing several biochemical and behavioral bookmarkers. The authors found oxidative damage on lipids and proteins as well as a synergism in the combination of 21oc and As exposure for small and lager invertebrates. This topic is pertinent given the need to assess the risk of metals such as As in a changing environment, namely warning changes. However, I found that several aspects required more development or key information. In addition to that, there are serious issues related to missing methodological information (QC/QA controls) that should be addressed. Considering these changes, I would recommend this article for publication in this journal.

R: The authors acknowledge the reviewer's comment. All comments were taken into consideration and the manuscript was revised accordingly.

Comment 1-The authors conducted a series of experiments including As exposure as well as temperature changes (16oC versus 21oC). Do you think the way your tested warming climate (from 16 to 21oC as described in lines 154-155) is a pertinent and realistic way to address this point? A temperature increasing daily 1oC is a realistic way to address global warming?

 R: The temperature of 16 °C was selected as representative of mean values measured at the sampling site, while the temperature of 21 °C was selected taking into account the future projections (IPCC 2018 estimated an average surface temperature increase between 2.6 °C and 4.8 °C by the end of this century [3] and Viceto et al. [4] indicated that the projected temperatures for Iberian Peninsula show an increase of over 6 °C by 2081–2100) (Lines 156-161).

The authors decided to increase gradually the temperature (1oC per day) following previous works [5,6]. However, the authors also read other works where temperature increase on tested organisms was performed in less time (e.g 2 oC per hour at Madeira et al., 2021), but in this case, these studies intended also to study the effects of heat waves, what was not our case.

Comment 2- Methodology section required further information to better evaluate the experiments performed:

 -Exposure experiments:

Do you measure As and other metals (Hg, Se) in food used during the nursery and exposure?

R: Thanks for your comment. In fact, we did not measure the content of As and other metals in the food used to feed polychaetes. However, we used commercial fish food, also used in other works [2,7], thus we believe that, if some trace elements, such as As, Hg, and Se are present in the food, it should be vestigial. Moreover, As concentrations in control organisms are very low, varying between 0.40±0.33 and 0.48±0.39 µg/g DW. But it was a good observation that it could be considered for future works.

Did As concentrations were stable during your experiments? Any stability test performed to check that? Did you make any comparison between nominal As concentrations versus As concentration in water? What about the sediment used? Did you check there was not any potential As contamination from the sediments? Did you perform any desorption and depuration analyses before measuring As in H. diversicolor?

R: The As quantification in test sediments used was previously quantified, and it was below the limit of detection. Regarding the stability of As concentrations, in previous experiments, it was determined the final concentration of As in water, without sediment and organisms, 7 days after the exposition and no significative differences were detected.  Also, no significant differences were found among these ranges of As concentrations in water and nominal As concentrations. Regarding As quantification in organisms, they were only washed in salt water after removing from the sediment but were not depurated.  

Metal measurements:

It is important to show your QC/QA control results. In section 2.3, you wrote that you used them but nothing is indicated later about these results. Any As contamination in blanks? What kind (source, biological sample) of certified reference material you used? How many replicates (n)? What is the detection limit for As? What does “bias error of the chemical analysis was less than 10% » mean?

R: Thank you for your observation. The requested information was included in the manuscript (lines 172-177). With the sentence “bias error of the chemical analysis was less than 10% “ we wanted to say that the coefficient of variation in tissue sample duplicates was less than 10%. In order to clarify, this sentence was substituted by “the coefficient of variation in tissue sample duplicates ranged from 4 to 8%”.

Biomarkers:

In the feeding activity measurement, it is not clear how the author estimated the time needed for your animals to detect and catch food? Any video recording analyses? How to distinguish detection and capture? References about this test should be added. Please provide more details about this endpoint since you obtained as results several seconds. How to ensure a good quality of such estimations?

 R: To estimate the time needed for animals to detect and catch food it was performed video recording analyses. For food detection, we considered the time that polychaetes needed to come to the sediment surface after adding the food to water; catching the food was considered the time that organisms needed to capture the food. More details were added to this endpoint. Nine polychaetes per condition were analysed (3 polychaetes per aquarium x 3 replicates). One reference was also added [2].

In section 2.5, it is necessary to better explain the different steps followed to obtain the samples for each biomarker. Even try to do a methodological scheme for each fraction or step. I think some improvements are needed and more importantly, to stay coherent with what you wrote in lines 189-207. In what fraction did you measure SOD and CAT? In the PMS fraction, you quantified proteins, sugars (as described in section 2.5.2) but that is not included at the beginning of section 2.5. FW indicates wet weight?

R: This section was improved as requested. FW indicates fresh weight and this information was included in the manuscript.

3- In this work, you generated sufficient data to get insights about the impact of As + temperature based on a combination of biochemical and behavioral endpoints, but you showed and discussed such results individually (biomarker by biomarker) without doing any global consideration, link, or correlation between these measurements. Why not prepare a PCA figure as an exploratory approach to distinguish similarities or differences in your results? To explore the link between the endpoints should be necessary. I was expecting something like that at the end of the manuscript.

R: The authors thank the reviewer's comment that it was very pertinent. In order to analyze if the global behavioural and biochemical response of H. diversicolor was influenced by the size, and temperature in the presence and absence of As, the data were submitted to an ordering analysis performed by Principal Coordinates (PCO), using the PRIMER 6 & PERMANOVA+. This analysis was included at the end of the results section (lines 470-484).

Minors comments.

-Title: Is it necessary to have a capital letter at the beginning of the word Arsenic?

 R: The authors changed “Arsenic” to “arsenic”.

-Abstract:

I really enjoyed reading the last sentences, but more development of this idea should be added in the discussion section.

 R: This idea was developed, as requested in the discussion section (lines 526-535).

-Keywords - the word “arsenic” is already in the title.

R: The word was removed and substituted by” Metalloids”.

-Introduction

Introduction section presents key information, but I was wondering if it is really needed in the first paragraph. Start your introduction talking about As, it is suitable to mention nothing about organic contamination (you did not address this point in your document).

 R: the first paragraph was removed as suggested.

Line 72: Are you saying as can show biomagnification? If yes, in what condition? What type of ecosystems?

R: This sentence was rewritten (lines 60-62)

 -Materials and Methods

Line 150: What was the stability of as in your experiments? Do you have a ratio of nominal As concentrations versus As measured in water?

R: Regarding the stability of As concentrations, in previous experiments, it was determined the final concentration of As in water, without sediment and organisms, 7 days after the exposition and no significative differences were detected. Also, no significant differences were found among these ranges of As concentrations in water and nominal As concentrations.

Line 157: Please show some metal concentrations already found in the area. Just to confirm you are working with environmentally realistic As concentrations.

R: More information about As concentrations in the environment was added (lines 152-154).

Line 181: At what temperature the PMS fractions were kept?

 R: Fractions were stored at -80ºC until further analysis. This information was added.

-Results:

In fig.1 you are showing metal concentrations in ug of As per weight of your organisms (g,  FW). Did you have any idea about the weight variation in your organisms during the 21 days of metal exposure? That should be interesting for results obtained for As exposure at 0,25 mg/L where As concentration decreased. Please, indicate the statistical test used in your comparisons. N=3 is a limited replicate number for some tests.

R: The statistical analysis was included (lines 239-246). By mistake, this part had not been included in the version submitted. Regarding polychaetes weight, we only recorded their weight at the end of the experiment, so we can have one idea about the weight variation if we compare it with the controls. The division among ages was done by their size.

In fig. 2:  it is not common that different life stages differed in their food capacity, even in their food type to ingest. So, why to make comparison between different class sizes? It should be more important to do that among As and T conditions than among class size.

 R:  The idea of the work was also to see which class size was more vulnerable when exposed to As, and since we measured the time that they need to detect and catch the food (this analysis was done with the same type of food), we decided to maintain the comparison among class sizes.

Discussion:

-Why not to discuss the results obtained in figure 1 for th4 three class sizes exposed to 0.25 mg/L?

R: These results were discussed (lines 508-514)

-The results regarding cholinesterase (ChE) activity required more development. How metal can affect the activity of this biomarker? More references to this point are expected to be added.

 R: More information about cholinesterase were included (lines 548-555).

-At times, the comparison of the results obtained is limited to organic contaminants, so why not to make more efforts in making these comparisons with other metals or elements?

R: More references with metals were included.

Round 2

Reviewer 2 Report

I really enjoyed the last version of the submitted manuscript where the authors addressed all the comments, suggestions, and critics I provided in my assessment rapport.

Now, I recommend accepting this manuscript.